# Gut Microbiota and Therapy in Metastatic Melanoma: Focus on MAPK Pathway Inhibition

**DOI:** 10.3390/ijms231911990

**Published:** 2022-10-09

**Authors:** Mora Guardamagna, Miguel-Angel Berciano-Guerrero, Beatriz Villaescusa-González, Elisabeth Perez-Ruiz, Javier Oliver, Rocío Lavado-Valenzuela, Antonio Rueda-Dominguez, Isabel Barragán, María Isabel Queipo-Ortuño

**Affiliations:** 1Medical Oncology Intercenter Unit, Regional and Virgen de la Victoria University Hospitals, Instituto de Investigación Biomédica de Málaga y Plataforma en Nanomedicina-IBIMA Plataforma BIONAND, 29010 Málaga, Spain; 2Department of Medicine and Dermatology, Medical School University of Málaga, Campus Teatinos, Blvr. Louis Pasteur, 32, 29010 Málaga, Spain; 3Medical Oncology Intercenter Unit, Group of Translational Research in Cancer Immunotherapy, Regional and Virgen de la Victoria University Hospitals, Instituto de Investigación Biomédica de Málaga y Plataforma en Nanomedicina-IBIMA Plataforma BIONAND, 29010 Málaga, Spain; 4Group of Pharmacoepigenetics, Department of Physiology and Pharmacology, Karolinska Institute, Tomtebodavägen 16, 171 65 Solna, Sweden; 5Department of Surgical Specialties, Biochemical and Immunology, Faculty of Medicine, University of Málaga, 29071 Málaga, Spain

**Keywords:** metastatic melanoma, gut microbiome, immune system

## Abstract

Gut microbiome (GM) and its either pro-tumorigenic or anti-tumorigenic role is intriguing and constitutes an evolving landscape in translational oncology. It has been suggested that these microorganisms may be involved in carcinogenesis, cancer treatment response and resistance, as well as predisposition to adverse effects. In melanoma patients, one of the most immunogenic cancers, immune checkpoint inhibitors (ICI) and MAPK-targeted therapy—BRAF/MEK inhibitors—have revolutionized prognosis, and the study of the microbiome as a modulating factor is thus appealing. Although BRAF/MEK inhibitors constitute one of the main backbones of treatment in melanoma, little is known about their impact on GM and how this might correlate with immune re-induction. On the contrary, ICI and their relationship to GM has become an interesting field of research due to the already-known impact of immunotherapy in modulating the immune system. Immune reprogramming in the tumor microenvironment has been established as one of the main targets of microbiome, since it can induce immunosuppressive phenotypes, promote inflammatory responses or conduct anti-tumor responses. As a result, ongoing clinical trials are evaluating the role of fecal microbiota transplant (FMT), as well as the impact of using dietary supplements, antibiotics and probiotics in the prediction of response to therapy. In this review, we provide an overview of GM’s link to cancer, its relationship with the immune system and how this may impact response to treatments in melanoma patients. We also discuss insights about novel therapeutic approaches including FMT, changes in diet and use of probiotics, prebiotics and symbiotics. Finally, we hypothesize on the possible pathways through which GM may impact anti-tumor efficacy in melanoma patients treated with targeted therapy, an appealing subject of which little is known.

## 1. Introduction

### 1.1. Epidemiology and Risk Factors of Cutaneous Melanoma

Cutaneous melanoma is a malignant proliferation of melanocytes with an increasing incidence over the last few years, probably because of higher sun exposure, longer survival in elderly patients and changes in lifestyle [1]. Before the era of immunotherapy and MAPK-directed therapy, melanoma was one of the deadliest cancers. However, a dramatic decline in mortality of nearly 30% has been seen since 2011, due to the appraisal of new therapeutic agents [2].

Ultraviolet radiation (RUV) is the main risk factor. As a result, melanoma constitutes a highly immunogenic tumor. The “UV mutational signature”, described in The Cancer Genome Atlas, is characterized by an increased number of translocations C>T in pyrimidines, which generates a high tumor mutational burden (TMB) [3,4]. The TMB is defined by the number of somatic mutations in the coding region of the genome or exome, and a high TMB results in an increasing number of neoantigens and immunogenicity of the tumor microenvironment (TME) [5]. The consequent inflammatory TME generated in this situation has proven to be an effective predictive biomarker for immunotherapy [6,7]. This has brought around immune checkpoint inhibitors (ICI) as successful therapies with nearly 40–60% objective response rates, a scenario previously unknown in metastatic melanoma [8,9,10]. Considering another major improvement in survival in melanoma, MAPK-directed therapies have also been under the spotlight over the last few decades. The oncogenic driver BRAF mutation, present in nearly 50% of melanomas—mostly young patients— has been described as inducing changes that facilitate tumor immune escape by the production of immunosuppressive cytokines, down regulation of Major Histocompatibility Complex I (MHC I) and recruitment of regulatory T cells (Tregs) and myeloid-derived suppressor cells (MDSCs) [11,12,13]. Thus, immune therapeutic agents and targeted BRAF inhibition therapy have become the mainstream in melanoma’s evolving landscape, as a higher amount of neoantigens present in these patients induces better responses to immune checkpoint inhibitors, and targeting BRAF pathway appears to have a favorable impact on immune surveillance in TME.

### 1.2. Gut and Skin Microbiome

The human microbiome is known to have an impact on cancer initiation, progression and response to treatments as well as modulation of the immune system and appearance of adverse effects [14]. However, although gut microbiota (GM) is being extensively studied and a link to cancer has already been established, the association of the skin microbiome, although potentially relevant, has not been extensively explored yet [15]. Therefore, in this review we will focus on gut microbiome. Table 1 summarizes the reported evidence on the association of gut and skin microbiome with cancer and drug response. 

### 1.3. The Influence of Gut Microbiota on the Response to Cancer Treatments

The GM is a huge group of germs that inhabit the human intestine from birth. Although it is still not well-known nor understood, GM has been linked to various diseases, including diabetes mellitus, metabolic syndrome, autoimmune diseases and cancer [32,33,34,35]. Gut Microbiota has also been reported as a predictive biomarker of response to chemotherapy, radiotherapy, immunotherapy and targeted therapy [36,37,38,39,40]. It is known to be modified by certain drugs; for example, in patients treated with citalopram that present an enrichment of Enterobacteriaceae, or those using antibiotics or proton-pump inhibitors where *Bacteroides* are predominant. At the same time, GM is fundamental for drug metabolism, reducing, decarboxylating, demethylating and deaminating components, influencing drug response and related toxicity. Indeed, GM-mediated deamination of 5-fluorocytosine to 5-fluorouracil increases the toxicity while reducing the affinity of the drug in colorectal patients [41,42,43]. A recent article published by Lung-Ngai Ting et al. reviewed the pharmaco–microbiomic interactions between microbiota, ICI and chemotherapy, highlighting the modulatory effect of GM in pharmacokinetics and pharmacodynamics, impacting on the efficacy and on the toxicity profiles [44]. This opens a completely new avenue in the characterization of the disruptors of efficiency and the determinants of drug toxicity.

### 1.4. Current Treatments in Metastatic Melanoma and Gut Microbiota

The availability of new treatments in melanoma and the better understanding of the underlying molecular mechanisms have made it easier to improve the management and quality of life of these patients. However, there is still much to gain as resistance to treatments eventually develops, a major problem that has not yet been solved. Gut Microbiota constitutes a current focus of research in the field of precision medicine. In relation to immunotherapy, GM has been described as playing a key role in immune homeostasis and in oncogenesis [45,46,47,48]. Indeed, response to ICI, development of resistance, as well as predisposition to immune-related adverse events (irAE) appear to be linked to the presence of certain bacteria in the gut [49]. This influence is being largely investigated in order to deepen the comprehension of the complex interaction between what seem to be key pathways in molecular oncology. While the association between ICI and microbiome has been well described, it remains to be assessed for MAPK-directed therapy (BRAF/MEK inhibitors—BRAF/MEKi), which is nowadays one of the key elements in melanoma treatment. This is worth highlighting, considering that nearly 50% of melanoma patients have *BRAF* mutations and can be treated with these drugs, benefiting from a 64–76% objective response rate and a 5-year overall survival rate of 31–35% [50,51,52]. However, it should be noted that these patients will eventually develop resistance to targeted therapy after 12–14 months of treatment, through either MAPK-dependent pathways, such as mutations in *NRAS* or *MEK*, or MAPK-independent pathways, such as upregulation of platelet-derived growth factor (PDGF) receptors or insulin growth factor (IGF) receptors in the PI3K-mTOR signaling pathway [53]. In view of this, unveiling new ways of increasing response rates and discovering new therapeutic targets becomes an urgent need. 

In this review, we will describe the interaction between GM, the immune system and BRAF/MEK-targeted therapy in the context of how they collectively impact anti-tumor efficacy. We will also analyze the relationship between ICI and microbiota, which may contribute to the yet-unstudied connection between GM and BRAF/MEKi. Our group is currently trying to define the role that gut microbiota may have in the outcome of metastatic melanoma BRAF-mutated patients treated with BRAF/MEKi.

## 2. Gut Microbiome and Cancer

The human microbiome is known to vary among individuals and also within a person over time, and has been described as influencing the progression of various diseases, from inflammatory bowel disease to major depressive disorders to cancer [54,55]. It is well known that some microorganisms in the human body can promote certain types of cancer, such as *Helicobacter pylori* in gastric cancer or *Human Papillomavirus* in cervical cancer [56,57]. Of note, preclinical models have shown that GM plays a fundamental role in initiation and progression of cancer, thus making it appealing as a therapeutic target that could change our daily clinical practice [58]. Changes in diet, use of antibiotics as well as other drugs, probiotics or lifestyle habits, impact the microbiome composition [36]. Therefore, the disease model has changed from a simplistic one to a more complex, intricate network of actors where the microbiome and the immune system may play a leading role in the response to treatments [59]. The influence of GM on cancer therapies, the immune system and carcinogenesis is illustrated in Figure 1.

### 2.1. Gut Microbiota and Carcinogenesis

The mechanisms by which GM may promote carcinogenesis have been proposed in numerous studies. A review published by Knippel et al. introduced these mechanisms: direct DNA damage by bacterial toxins, inflammation due to interaction between host cells and chronic infections, generation of metabolites and the inhibition of anti-tumoral immune responses [59]. An example of this is the secretion of a toxin by *Bacteroides fragilis* that induces the cleavage of E-cadherin, which in turn favors transcription by translocating ß-catenin to the nucleus and causes colonic hyperplasia. *Fusobacterium nucleatum* also acts in this pathway by linking E-cadherin to an adhesin called FadA, releasing ß-catenin [60]. This state of chronic inflammation exerted by some microorganisms and genotoxic toxins, including cytolethal distending toxin produced by Gram-negative bacteria and colibactin released by Enterobacteriaceae, favors tumorigenesis by mediated genotoxicity with reactive oxygen species, direct DNA damage and genomic instability [61]. The metabolism of some nutrients undergone by gut bacteria can also generate carcinogenic products such as aromatic amines, acetaldehyde or sulfide [62].

The tumor whose link to microbiome is best established is colorectal cancer (CRC), probably due to its high incidence and added to the fact that dietary habits have a well-known correlation with colon carcinogenesis and GM composition. The mechanisms proposed to participate in colorectal carcinogenesis are the alteration of the intestinal epithelial barrier, the production of genotoxins and the secretion of toxic metabolites by pathogenic bacteria [63]. Herbert et al. introduced the idea that microbial interventions could be beneficial in CRC, and although there are studies where it seems to be a plausible method to increase anti-tumor effects, there is still an urgent need to clarify its benefit in clinical models and daily practice [64,65]. 

As regards other tumors, microbiome has also been proposed to play a multi-factorial role in breast cancer: modulating metabolism of chemotherapeutic drugs, regulating tumor initiation and progression, and influencing therapy response and resistance [47]. In endometrial cancer it has been suggested that it may be involved in the body response to treatment and the presence of adverse events [66]. In preclinical mice models of pancreatic cancer, it was seen that *Escherichia coli* and *Enterococcus* introduced orally migrated to the pancreas and modified its microenvironment, and those animals who were treated with antibiotics also had a higher number of migrated bacteria in TME [67]. 

### 2.2. Evidence of Gut Microbiota and Melanoma Carcinogenesis

Concerning melanoma, Mekadim et al. identified associations between dysbiosis in skin microbiota, GM and melanoma progression in porcine models [14]. Changes in gut microbiota from early stages to metastatic disease were described by Vitali et al., who observed an abundance of *Prevotella copri* and yeasts of the *Saccharomycetales* species in melanoma patients compared to controls [68]. A review recently published by Makaranka et al. managed to describe the current evidence regarding the carcinogenic and anti-tumoral properties of GM in melanoma, emphasizing its connection with response to radiotherapy, chemotherapy and ICI and the impact that diet and probiotics may have in this scenario [69]. As an example, GM has been reported to trigger the immune system to strengthen the abscopal effect caused by radiotherapy, apart from promoting radiotherapy-adverse effects such as mucositis, colitis and bone marrow failure [70]. Moreover, chemotherapeutic agents, such as cyclophosphamide and cisplatin, appear to alter the composition of GM by stimulating the translocation of Gram-positive bacteria into secondary lymphoid organs, with the consequent induction of pathogenic Th17 cells and memory Th1 responses. Indeed, the use of antibiotics has been associated in preclinical mice models with resistance to cyclophosphamide [71]. Cisplatin efficacy also appeared to be reduced by antibiotic administration by a similar mechanism involving Th17 cells and oxygen-reactive species [19].

## 3. Gut Microbiome and Immune System

Nearly 4 × 10^13^ microbial cells are present in the human body, and almost 95% of these live in the gut. The immune-regulating role of GM and the influence of the GM and dysbiosis in disease and treatment outcomes are the predominant objects of studies in the GM field [15]. An example of this was described in preclinical models by Sivan et al., who found that mice had different anti-tumor immune responses depending on their microbiome [19]. Also, the modulation by GM of CD8+ T cells, T helper 1 (Th1) and tumor-associated myeloid cells has been evidenced in preclinical mice models, where antibiotics and the absence of microorganisms exerted a negative effect on ICI response [72]. On the other hand, it was also reported that certain bacteria acting on innate and adaptive immunity may cause a favorable microenvironment for tumor initiation and progression [73]. This shows the complex interconnection between GM and immunity, which comprises macrophages, monocytes, dendritic cells, antimicrobial peptides and lymphoid cells, among others [74]. This duality is what makes it more intriguing when trying to apply GM on therapeutics. 

### 3.1. Innate Immunity

Innate immunity is the first barrier that confronts external threats, being that the intestine is one of its main players. It has been described that breast milk containing bacteria such as *Lactobacillus* and *Bifidobacterium* may have a crucial role in the development of this barrier in neonates [75]. Moreover, GM is known to modify the host’s response to vaccine immunity. This is of particular importance as gut dysbiosis is impaired in the elderly, which predisposes them to various diseases, a key point to take into consideration in the current COVID19 pandemic as the immunity provided by the vaccination may also be diminished in this subgroup of patients [76]. 

It is interesting to highlight the immunological footprints on which all this knowledge is based, being that the Complement is one of the key players in innate immunity. The Complement includes three pathways: the classical, the alternative and the mannose-binding lectin (MBL), which are activated either by antibody-dependent or independent mechanisms, such as recognition of certain components of the walls of microorganisms by C1, C3 and MBL. These pathways converge in the cleavage of C3 into C3a and C3b, which ultimately results in the formation of the membrane attack complex leading to cell lysis. This mechanism, which has been known for more than a century, was initially classified as an effector against tumor growth due to the fact that modifications in tumor cell membranes are recognized and triggered by the immune response [77]. However, there is a controversy about the final outcome of its activation. Taking into account the effects of chronic inflammation on the tumor microenvironment, new hypotheses have been developed in the search for complement inhibitor therapies, breaking apart the current search for activation of these pathways [78,79,80]. 

Pio et al. already described in 2013 that the Complement might exacerbate chronic inflammation, thus stimulating an immunosuppressive microenvironment, apart from inducing angiogenesis and cancer-promoting signaling pathways [77]. Markiewski et al. verified the role of C5a in tumor growth [81]. On the other hand, Bulla et al. also demonstrated that C1q deficiency in mice correlated with a lower probability of tumor progression and metastasis compared to wild-type mice [82]. In the wake of these findings, many studies discuss the need to shed light on the relationship between the Complement and the tumor microenvironment [79,83,84]. However, they all emphasize that these findings are part of the tumor microenvironment, and not part of the tumor cells themselves. This reinforces the importance not only of mutations but of a microenvironment that favors tumor development and progression.

Moreover, it is also worth highlighting the key role of neutrophils, main actors in innate immunity, mediating a suppressive immune response in TME. This is accomplished by the generation of reactive oxygen species, the secretion of immunosuppressive cytokines, the formation of neutrophil extracellular traps involved in cancer migration and invasion and the expression of PDL1, among others, all of which lead to immune tolerance and T lymphocyte anergy [85,86].

Once the link between innate immunity and cancer has been reviewed, it seems interesting to highlight a few studies evaluating the impact of GM on complement activation in other daily-life scenarios, such as the induction of preterm birth or neovascular age-related macular degeneration [87,88]. Chehoud et al. demonstrated that variations in the skin microbiota alter complement gene expression and vice versa by associating C5a receptor signaling to the diversity and composition of skin microbiota [89]. There are not many more studies on the subject, which highlights the need for further research on the impact of microbiome and cancer.

### 3.2. Adaptive Immunity

Adaptive immunity, as described before, also appears to be modified by the GM, by interacting with dendritic cells, macrophages and lymphocytes. It is worth mentioning the induction and diversification of B cells, the production of antibodies, mainly IgA, the influence on the response of CD4+ cells and the secretion of pro-inflammatory cytokines [90,91]. In a study of pancreatic cancer and microbiome, it was shown that removal of dysbiotic and immune-suppressive GM, such as bacteria from genera Firmicutes, Proteobacteria, Actinobacteria or Bacteroidetes, generated high levels of TNFα and IFNγ expression potentiating Th1 CD4+ cells and cytotoxic CD8+ T cells and increasing the anti-tumoral response [92]. This highlights the oncogenic role that certain bacteria may have in cancer.

### 3.3. Gut Microbiota, Chronic Inflammation and Effects over Tumor Microenvironment

Additionally, GM’s production of metabolites, for example butyrate and propionate, can stimulate the release of certain cytokines such as IL-6 or TNFα. As a result, immunosuppressive Tregs are stimulated, which favors an inflamed TME. Chronic inflammation has long been described as contributing to initiation and progression of cancer. This is a result of induced mutations, avoidance of apoptosis in cells, stimulation of angiogenesis and generation of adaptive responses, all of which confer tumor cells a survival advantage [47,93,94]. The immune reprogramming in TME has been established as one of the main targets of the microbiome, and not only by promoting these inflammatory responses, but also by inducing immunosuppressive phenotypes or even conducting anti-tumor responses. As a result, ongoing clinical trials are evaluating the role of fecal microbiota transplant (FMT), dietary supplements and impact of use of antibiotics and probiotics as predictors of response to ICI and other treatments [36,92]. It would also be appealing to understand the result of these interventions in patients treated with MAPK inhibitors to enhance therapeutic approaches. 

In summary, although GM is already known to modulate innate and adaptive immunity, disentangling the specific mechanisms that link the alterations of these immunities and the drug response rates is a major need for the development of GM-based therapeutic strategies.

## 4. Gut Microbiome and Immune Checkpoint Inhibitors

GM and its immune-modulating role in patients treated with ICI has been and continues to be extensively studied. It has been reported that patients receiving antibiotics prior to or during treatment with ICI have a worse prognosis with a decrease in progression-free survival and overall survival. On the contrary, FMT may improve treatment response [36]. However, it must be highlighted that the techniques used for transplantation may transfer immunosuppressive and pathogenic microbes, so the effectiveness of these procedures needs to be studied in depth and validated [95].

### 4.1. Gut Microbiota Influencing Response to Immune Checkpoint Inhibitors

A previous study reported that patients who have a GM rich in Clostridiales, Ruminococcaceae or *Faecalibacterium* have higher levels of CD4+ and CD8+ T cells thus generating a better response to immunotherapy [29]. On the other hand, predominance of Bacteroidales in the gut is related to a majority of Tregs and myeloid-derived suppressor cells (MDSC), with poorer response to ICI [14]. Sivan et al. reported the *Bifidobacterium’s* association to antitumor activity by augmented dendritic cell function and activation of CD8+ T cell response in the tumor microenvironment (TME) [19]. These findings support the idea of the modulating role of GM in patients treated with ICI.

Roviello et al. reviewed the impact of GM on the efficacy of ICI, associating certain microorganisms of the phylum Firmicutes (Clostridiales, *Enterococcus faecium*, *Faecalibacterium prausnitzii*, *Gemmiger formicilis*, *Lactobacilllus*, *Ruminococcus*), Bacteroidetes (*Alistipes*, *Bacteroides caccae*), Actinobacteria (*Collinsella aerofaciens*, *Bifidobacterium longum*), Proteabacteria (*Klebsiella pneumoniae*) and Verrucomicrobia (*Akkermansia muciniphila*) with immune checkpoint responders, and, on the other hand, Firmicutes (*Faecalibacterium prausnitzii*, *Roseburia intestinalis*, *Ruminococcus obeum*), Bacteroidetes (*Bacteroides thetaiotaomicron*, *Parabacteroides distasonis*) and Proteabacteria (*Escherichia coli*) with non-responder patients [96].

### 4.2. Gut Microbiota and Immune Checkpoint Inhibitors in Melanoma

Baruch et al. conducted a clinical trial to evaluate FMT and re-induction of anti-PD1 immunotherapy in 10 melanoma patients who were refractory to ICI, and demonstrated a clinical response in three of those patients [97,98]. These may be suggestive of a modulation of the immune system by the GM, but uncertainty exists on whether that clinical response could be due to a delayed reaction to previous anti-PD1 therapies [36]. Based on these findings, there were several groups who examined a potential association between GM and clinical response to ICIs, demonstrating different microbiota composition between responders and non-responders [37]. An example of this was a study carried out on 42 stool samples taken from metastatic melanoma patients before immunotherapy treatment, which showed that bacterial species in responders included *Bifidobacterium longum*, *Collinsella aerofaciens*, and *Enterococcus faecium*, whereas non responders had a predominance of *Ruminococcus obeum* and *Roseburia intestinalis*. The responder profile was also associated with an increase in Batf3-lineage dendritic cells and Th1, evidencing immune activation and supporting the hypothesis that responder-associated bacteria may have an impact on innate and adaptive immunity [44].

Kumar et al. published a review summarizing the impact of microbiome on response to immunotherapy, apart from discussing the role of dietary habits on melanoma progression and treatment. They described that certain germs including *Actinobacteria* spp., *Bacteroides thetaiotaomicron* and *massiliensis*, *Bifidobacterium adolescentis*, *longum* and *pseudolongum*, *Enterococcus faecium*, *Faecalibacterium prausnitzii*, *Lactobacillus* spp., *Klebsiella pneumoniae*, *Parabacteroides merdae*, had beneficial effects on melanoma, while Bacteroidales, *Roseburia intestinalis*, *Escherichia coli*, *Actynomyces odontolyticus*, Peptostreptococcaceae and Proteobacteria had a negative impact on melanoma’s clinical evolution [15]. 

It is worth noting that although the microbiome composition is beginning to be seen as a predictive biomarker for immunotherapy, a combination of multiple biomarkers could be better to predict the efficacy of this treatment, as for now none of the biomarkers studied—PDL1, tumor mutational burden, tumor infiltrating lymphocytes—possesses high enough sensitivity and specificity to predict response rate [99]. 

To summarize, the increasing evidence of the correlation of the immunotherapy response with the presence of certain bacteria in the gut leads us to think that they are potential predictive biomarkers. However, the fact that the response in different types of cancer was associated to different bacteria, as well as contrasting responses to treatment depending on the context, reveals a complex connection that still needs to be uncovered.

## 5. Mapk Therapy and Immune System

The MAPK pathway, shown in Figure 2, is the result of a kinases cascade activation that, after several phosphorylations, leads to cell proliferation and survival through growth factors. The kinases RAF, MEK and ERK are also known as MAPKKK, MAPKK and MAPK because of their function as enzymes that transfer phosphate groups. The MAPK pathway plays an essential role in promoting TME inflammation and evading the immune system, as a result of paracrine and autocrine secretion of tumor growth factors and cytokines, apart from maintenance of proliferation and reduced apoptosis in cancer cells [100]. The most studied molecule of the pathway, KRAS, was found to cause immunosuppression by several mechanisms, for example upregulation of PDL1, infiltration of MDSCs in TME, stimulation of Tregs by secretion of IL-10 and TGF-β1, or downregulation of MHC-I, which impairs the recognition of tumor associated antigens and neoantigens by CD8+ T cells [101,102,103]. Moreover, the alteration of signaling between cells induced by this pathway through impaired CD40:CD40L blocks maturation of antigen- presenting cells and activation of CD8+ T cells [104].

### 5.1. BRAF/MEK Inhibition and Tumor Microenvironment

Once considering the immunosuppressive effects of the MAPK pathway, it can be hypothesized that BRAF/MEK inhibition, although not designed to target the immune system, influences the tumor-immune microenvironment and anti-tumor immune responses [105]. This has already been described in the literature, and justifies the synergic response seen when combining ICI and BRAF/MEKi [106]. Trojaniello et al. concluded that the co-administration of these drugs could induce tumor regression as well as prolong the immune response thanks to ICI [107]. Consistently, Devji et al. demonstrated an increase in overall survival without implying an important number of adverse effects [108]. There were various studies that also confirmed promising results of this combination, but taking into account an increase in “manageable” adverse reactions [109,110,111,112]. 

Another important point to take into account is the action of targeted therapy in TME. Several studies have also demonstrated that increasing T cell infiltrates in TME appear in patients with metastatic melanoma treated with BRAF/MEKi, enhancing a favorable anti-tumor microenvironment and decreasing immunosuppressive markers, which implies a higher survival rate [113,114,115,116]. Even though an intact immune system is essential for a good response to directed therapy, T lymphocytes in TME are critical to maintain the therapeutic effects of BRAF/MEK inhibitors [117]. 

### 5.2. BRAF/MEK Inhibition and Immunity

Focusing on BRAF/MEKi and their effects on immunity, a review by Kuske et al. exposed that BRAF inhibition caused a paradoxical activation of the MAPK pathway by increased phosphorylation of ERK in CD4+ and CD8+ T cells thus potentiating the immune response, while MEK inhibition apparently did not influence lymphocyte functions in vivo [117]. Erkes et al. agreed that BRAF inhibition also increases MHC expression and CD4+ and CD8+ T cell-response, but highlighted that MEK inhibition appears to act on T cell receptor (TCR)-mediated apoptosis [118]. Overall, BRAF inhibition seems to have a well-established anti-tumoral function, and while MEK seems to have activity over T cell function in vitro, controversy is seen when using in vivo models.

When considering targeted therapy’s action over the immune system, it has been described as increasing CD8+ T cell infiltrates and antigen expression, as well as decreasing immunosuppressive cytokines. Markers of immune exhaustion, such as PD-1 or TIM-1, appear to be upregulated as a mechanism that modulates the immune system even before resistance to iBRAF/MEK appears [119]. During treatment in preclinical and clinical models, PDL-1 is also upregulated, suggesting an acquired resistance to BRAF/MEK inhibition [113,120]. Interestingly, Berciano et al. have described how targeted therapy can induce immune re-induction by measuring changes in expression of genes involved in immune response. The fact that there are genes that may play a role in maintaining response to BRAF/MEK inhibitors, and others that are involved in resistance at progression by escaping immune surveillance, may help us understand the intrinsic pathways between immune system, targeted therapy and GM. Of mention, genes such as C-X-C Motif Chemokine Ligand 10 (CXCL10) and Serpin Family G Member 1 (SERPING1), regulators of T cells and classical complement pathway, respectively, appear to be upregulated in metastatic melanoma treated with iBRAF/MEK, thus suggesting an immunomodulatory role of these genes which may postulate them as possible therapeutic targets to enhance response to targeted therapy [121]. It is worth mentioning that as there are genes of immune re-induction and markers of immune exhaustion expressed during iBRAF/MEK treatment, combination therapy with ICI seems an appealing setting, and it has already begun to show interesting synergic responses with better progression-free survival and durability of response, yet with increasing levels of toxicity [122]. Other combinations of immunotherapeutic drugs and iBRAF/MEK should be explored, considering the entangled bond between these agents, microbiome and the immune system. 

Overall, the activity of iBRAF/MEK in immunity and TME seems to play an interesting role in response to treatment, which may help us enhance effectiveness of targeted therapy and search for new pathways that can be triggered to reduce the development of resistance.

## 6. Gut Microbiome and Mapk Inhibitors

Although the association between GM and BRAF/MEKi has not been described, it is an interesting field of research that needs to be unfolded, considering that it represents one of the main therapies in melanoma patients and that it may help us have a better understanding of molecular pathways, the immune system and microbiome. The outcomes that changes in diet and use of antibiotics, probiotics or FMT may lead to in carcinogenesis and response to therapy may help us deepen our comprehension of how to approach patients with BRAF-mutated melanoma. Apart from this, analyzing GM in cancer patients may determine responders and non-responders, so as to enhance other strategies. Defining the constitution of the GM could become an easier and more innocuous procedure if confirmed as a predictive or prognostic biomarker. 

When focusing on the relationship between MAPK inhibitors and GM, although it has not yet been described in melanoma patients, it has already been studied in other types of cancer. Trivieri et al. analyzed fecal samples of 33 colorectal cancer (CRC) patients classified in BRAF-mutated or wild-type, and 13 healthy subjects. Patients with BRAF mutations were characterized by an abundance of *Fusobacteria*, *Prevotella enoteca*, *Prevotella dentalis*, *Hungateiclostridium saccincola*, *Sutterella megalosphaeroides*, *Stenotrophomonas maltophilia* and *Victivallales bacterium*. Interestingly, they also analyzed xenogeneic BRAF-mutated CRC models, which showed a higher microbial diversity than BRAF wild-type CRC controls. The BRAF-mutated microbiota signature was closer to healthy controls than the wild-type CRC model. The reason why patients with this mutation present with a more eubiotic condition is still not known, but it has been hypothesized that gut dysbiosis may have an influence only in the development of conventional CRC, or that fewer microorganisms in BRAF-mutated carriers are sufficient to initiate an oncogenic pathway [123]. Other studies showed that *Porphyromonas gingivalis* promotes proliferation of CRC in vitro by the activation of the MAPK pathway, and that *Leuconostoc mesenteroides* promotes apoptosis by modulating NF-κB, AKT, PTEN and MAPK [124,125]. Figure 3 illustrates the relationship between the MAPK pathway, the immune system and gut microbiota.AQUI.

## 7. Gut Microbiome and Interventions

Dietary interventions, such as prebiotics, probiotics, symbiotics or the use of antibiotics as a way of modulating GM have gained interest in the past few years, due to the increased knowledge acquired about the relationship between GM and cancer. 

### 7.1. Prebiotics, Probiotics and Symbiotics

Prebiotics are substances present in food that confer a health advantage to the host by modulating microbiota, mainly stimulating the growth of beneficial bacteria [126]. The most studied include fructooligosaccharides and inulin [127]. It seems that these components stimulate differentiation of colonic cells, thus inhibiting the formation of pre-neoplastic lesions [128]. 

Probiotics are living organisms that provide a beneficial outcome when administered, and contribute to a better microbial balance [129]. Examples of these are *Lactobacillus* and *Bifidobacterium*, commonly used as probiotics. It has been described that apart from maintaining homeostasis they can modulate response to treatment, such as *Clostridium butyricum*, which was seen to potentiate the efficacy of nivolumab and ipilimumab in patients with kidney cancer [130]. Moreover, it appears that they may induce better tolerance by reducing adverse effects, as reported by Wang et al. with the administration of *Bifidobacterium* in mice [131].

Symbiotics are combinations of prebiotics and probiotics. The use of this interesting combination appears to be more effective than the monotherapy, and it is already being studied in depth. Dos Santos Cruz et al. found that microbiota had a better antiproliferative and anti-carcinogenic function when using symbiotics compared to probiotics on a colorectal carcinogenesis model [132]. Dey et al. described the reduced incidence of adverse effects of chemotherapy and the antitumoral impact of symbiotic formulations administered in humans [133].

### 7.2. Fecal Microbiota Transplantation

Another interesting approach that has several active clinical trials ongoing is the FMT; considering only melanoma patients, these include NCT04577729, NCT04988841, NCT05251389 and NCT03819296 [134]. Fecal Microbiota Transplantation involves removing fecal matter from a donor and transplanting it into the bowel of a patient in order to provide a health benefit, and it is already being used in recurrent *Clostridium difficile* infections, as well as being tested in other diseases such as irritable bowel syndrome, metabolic disorders and cancer [135]. Gopalakrishnan et al. described that mice who had FMT from anti-PD1 melanoma responders, which were abundant in *Faecalibacterium*, had a decrease in tumor growth and this correlated with an increase in CD8+ T cells and innate immune cells in TME. On the other hand, mice with FMT from non-responders had an abundance of Th17 cells and Tregs, suggesting an immunosuppressive response [29]. Apart from modulating the immune system, it has been described that FMT can increase survival rates in mice with cancer undergoing radiation therapy, as well as diminishing adverse effects from anti-cancer therapies such as chemotherapies or tyrosine kinase inhibitors [136]. 

### 7.3. Dietary Habits

Although GM remains quite stable during life, there are certain factors that can modulate it, such as diseases, dietary habits and the environment, which may predispose the host not only to cancer but also to metabolic diseases such as diabetes and obesity [16,137,138]. A Mediterranean diet, compared to a high-calorie Western diet, impacts on the gut microbiome and on the host’s health. It is known to increase beneficial bacteria, including *Lactobacillus* or *Faecalibacterium*, stimulating an anti-inflammatory environment and reducing oxidative stress [139]. Moreover, nutrients included in this type of diet have an antioxidant effect and exert a protective, anti-proliferative action, and have been found to decrease certain types of cancer such as breast, gastric, upper digestive and respiratory tract cancers [140].

### 7.4. Antibiotics

Antibiotics also appear to have a role in this scenario. Although they have long been and are still being used for cancer treatment—examples include adriamycin, epirubicin or bleomycin—in certain cases they may have a pro-tumoral effect by stimulating chronic inflammation, inducing genotoxicity and diminishing immune response [141]. In addition, antibiotics disrupt GM that can have an impact on adverse effects due to reduced metabolism of treatments in the gut, or even reduce the efficacy of certain chemotherapies by reducing cytotoxic T cell response [133].

In summary, treatment possibilities in cancer seem to imply not only cytotoxic agents, immunotherapy and targeted therapies, but also interventions in dietary habits. However, there is still much to gain on this subject as evidence, although increasing, is still scarce.

## 8. Conclusions

Gut microbiome has become an interesting subject in the past few years due to the increasing knowledge of its impact on immunity, carcinogenesis and response to treatments. Its role as a “modulator of immunity” makes it a plausible option for searching for new therapeutic strategies, considering that tumor cells lead to an immunosuppressive phenotype. This can be enhanced with first-line treatments already used in clinical practice in metastatic melanoma, but the fact that resistance develops in most patients makes it challenging to find new pathways that can overturn this. The most interesting evidence sheds light on the link between responders to ICI and certain bacteria, thus reinforcing the idea that certain interventions and even FMT might seem appealing to potentiate favorable results in patients. On the other hand, although the association between BRAF/MEKi and GM in melanoma has not been described in previous studies, the role that both have in modulating the immune system demonstrates indirect signs of a possible correlation between them. As a result, the study of the composition of the GM in these patients could help us understand and probably predict responses to targeted therapy. This brings up the hypothesis that the GM may be used as a predictive biomarker of response to targeted therapy, considering the available evidence that establishes a link between microbiome and response and toxicity to ICI and also to other drugs in pharmacokinetic and pharmacodynamic terms. Future studies are pending to confirm this association and clarify the landscape in this matter.

In conclusion, this review has summarized the available evidence on the association of GM and immunity in the context of metastatic melanoma’s first-line treatments. The impact of ICI and targeted therapy on the immune system and its link to the gut microbiome brings to light pathways involved in carcinogenesis and resistance to treatments, as well as revealing key actors that play a fundamental role in responders. Identifying the bacteria that influence response to treatments may provide new strategies that enhance the approach to melanoma patients. This could be useful for identifying new predictive and prognostic biomarkers that could contribute to a better prognosis of these patients. Furthermore, melanoma being one of the most immunogenic tumors due to its high TMB makes the immune system an appealing target to work on, and ICI have already proven a point in this scenario. Future perspectives will undoubtedly focus on precision medicine, highlighting the role of the immune system and molecular pathways in cancer, as oncology has become a multifactorial disease that is not only influenced by tumor cells and their microenvironment, but also by host factors such as microbiome, received treatments, dietary habits and other external and internal components of which there is still much to unveil.

## Figures and Tables

**Figure 1 ijms-23-11990-f001:**
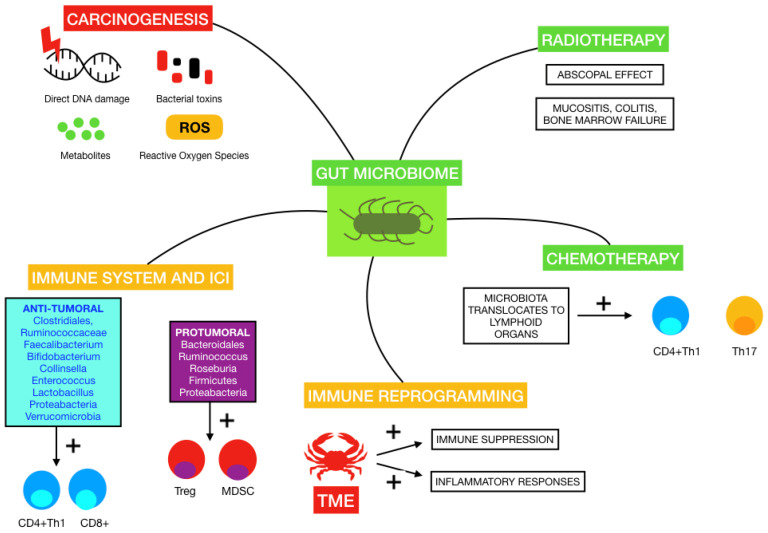
Influence of some of the mentioned gut microbiomes on carcinogenesis, immune system and response to immune-checkpoint inhibitors (ICI), immune reprogramming on tumor microenvironment (TME), radiotherapy and chemotherapy.

**Figure 2 ijms-23-11990-f002:**
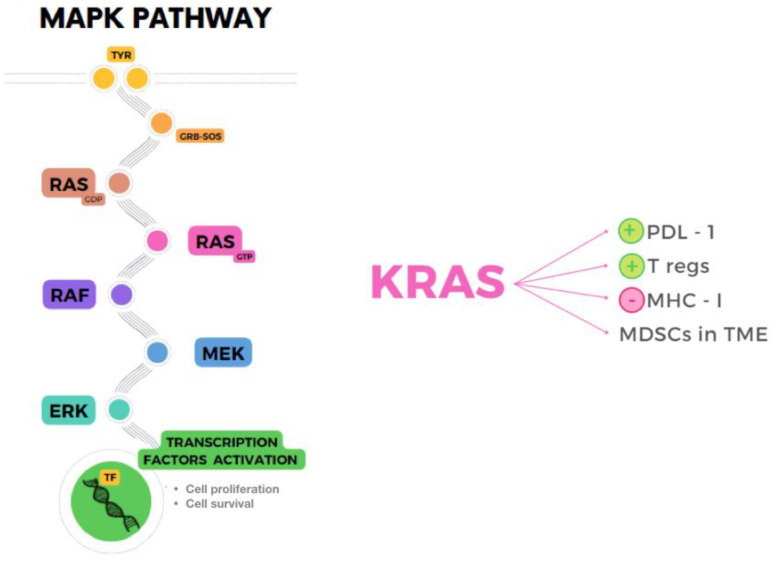
MAPK pathway (**left**) and immunosuppressive role of KRAS (**right**) mediated by upregulation of PDL-1, increased Tregs and MDSCs in TME, and downregulation of MHC-I.

**Figure 3 ijms-23-11990-f003:**
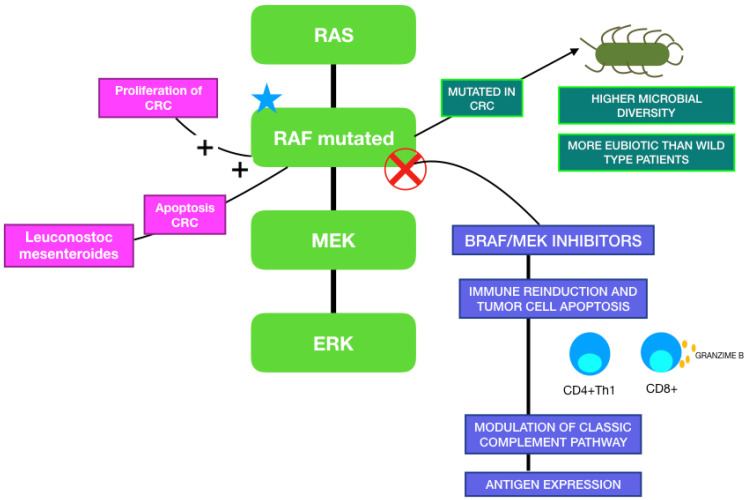
MAPK pathway and its correlation with gut microbiome and BRAF/MEK-targeted therapy.

**Table 1 ijms-23-11990-t001:** Evidence of the impact of skin and gut microbiome in cancer and response to treatment, and associated biomarkers. PFS: progression-free survival, OS: overall survival.

Microbiome	Reference	Bacteria	Results	Biomarker
**Skin**	Mizuhashi, S., et al. [16]	*Corynebacterium*	Advanced stages (III/IV) acral melanoma	IL-17A
	Naik, S., et al. [17]	*Staphylococcous epidermidis*	Normalizes IL-17A production, related with tumor growth and anti-tumor immunity	IL-17A
	Nakatsuji, T., et al. [18]	*Staphylococcous epidermidis*	Reduces the incidence of UV-induced skin tumors	6-HAP (6-N-hydroxyaminopurine)
**Gut**	Sivan, A., et al. [19]	*Bifidobacterium*	Enhances anti-tumor response of anti PD-1	CD8+ T cells
	Bessell, C.A., et al. [20]	*Bifidobacterium*	Enhances anti-tumor immunity by amplifying T cells	CD8+ T cell epitope SVY
	Vétizou, M., et al. [21]	*Bacteroides fragilis*, *B. thetaiotaomicron* and, *Burkholderia*	Associated with response to anti-CTLA4	IL-12 induced T cell response
	Miller, P.L., Carson, T.L. [22]	*Bacteroides fragilis*, *Burkholderia cepacia* and *Faecalibacterium*	Associated with response to anti-CTLA4	IL-12 induced T cell response
	Frankel, A.E., et al. [23]	*Bacteroides caccae*, *Streptococcus parasanguinis*, *Faecalibacterium prausnitzii*, *Holdemania filiformis*, *Bacteroides thetaiotamicron and Dorea formicigenerans*	Associated with response to immune checkpoint blockade	Anacardic acid and other metabolites
	Wind, T.T., et al. [24]	*Streptococcus parasanguinis* and *Bacteroides massiliensis*	Associated with PFS and OS, respectively, in response to immune checkpoint blockade	Aspartate, thiamine diphosphate, NAD/NADH, glycolysis, TCA and glyoxylate, and pyruvate pathways
		Peptostreptococcaceae	Shorter PFS and OS	Peptidoglycan and methanogenesis pathways
	Chaput, N., et al. [25]	*Faecalibacterium* and Firmicutes	Longer OS, PFS and immune-induced colitis when treated with anti-CTLA4	CD4+ T cells and higher increase in serum CD25 cells
	Tanoue, T., et al. [26]	Bacteroides, Ruminococcaceae, *Fusobacterium*, *Phascolarctobacterium*, *Eubacterium*, *Paraprevotella*, *Alistipes*	Enhanced efficacy of ICI	Interferon-γ-producing CD8 T cells
	Mager, L.F., et al. [27]	*Bifidobacterium pseudolongum*, *Lactobacillus johnsonii*, and *Olsenella*	Enhanced efficacy of ICI	Increased inosine and anti-tumor T cells
	Matson, V., et al. [28]	*Bifidobacterium longum*, *Collinsella aerofaciens*, and *Enterococcus faecium*	Enhanced efficacy of ICI	SIY–specific CD8+ T cells
	Gopalakrishnan, V., et al. [29]	Ruminococcaceae and Clostridiales	Responders to anti-PD-1	CD45+ and CD8+ immune T cells
		Bacteroidales	Non responders to anti-PD-1	RORγT+ Th17, CD4+ FoxP3+ T cells, CD4+ IL-17+
	McCulloch, J.A., et al. [30]	Actinobacteria, *Lachnospiraceae*, *Ruminococcaceae*	Responders to anti-PD-1	Protective membrane mucins (MUC13 and MUC20) and apolipoproteins (APOA1, APOA4 and APOB)
		Bacteroidetes and Proteobacteria	Non responders to anti-PD-1	High neutrophil–lymphocyte ratio and proinflammatory cytoquines (ILB, CXCL8, SOD2)
	Limeta, A., et al. [31]	*Faecalibacterium and Barnesiella intestinihominis*	Responders to anti-PD-1	Upregulation of inositol metabolism and vitamin B pathway
		*Bacteroides*	Non responders to anti-PD-1	Upregulation of biosynthesis pathways

## Data Availability

Not applicable.

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
