# Peer review of "Gut Microbiota and Therapy in Metastatic Melanoma: Focus on MAPK Pathway Inhibition"

_ijms, 2022, doi:10.3390/ijms231911990_

Round 1

Reviewer 1 Report

In this review Guardamagna et al, discussed the link between gut microbiota and cancer, its anti-tumor effect by targeted therapy majorly focused on MAPK pathway inhibition. Author explained every aspect of the review very clearly and in detail with the help of required figures and table. I do not have any major comment except that author should mention the detailed conclusion in end of the study and proofread the manuscript very carefully.

Reviewer 2 Report

 The review article deals with rather complex interrelations between gut mictobiota and immune mechanisms of malignancy control. The effects of two different types of immune-targeted drugs, immune checkpoint inhibitors (ICI) and BRAF/MEK inhibitors are highlighted, because of their potential usage for melanoma treatment.

As main item, the role of gut (to lesser extent, skin) microbiomes is discussed, due to its major role in development of the overall antitumor response. The authors suggest dual effects of microbiota, i.e., toxicity and carcinogenicity of some gut bacteria, and, generally, positive effects on antitumor immunity caused by the microbiota, including potential promotion of chemotherapy efficiency.  The main point concerns probable mechanisms of immune therapy by ICI based on clinical data on immunomodulatory effects of Bacteroides and other bacterial classes. Probable manipulation of microbiota by pre- and probiotic therapy or fecal microbiota transplantation (FMT) is considered a promising way to promote antitumor immunity, improve microenviroment and modify clinical responses to ICI therapy in metastatic melanoma. However, further experimental and clinical data are required in the field.

Hence, the article provides certain information concerning interactions between gut microbiome, and ICI, or BRAF-targeted drugs in melanoma patients.

Remarks:

1)     The section Innate Immunity (Lane 211 and further on), concerns the effects of complement activation as a potentially carcinogenic factor in chronic inflammation. In this view, the role of granulocytes as ROI producers could be also mentioned, due to their presence in the inflammatory foci.

2)     Unlike oral or colon cancer, the role of gut microbiota-induced changes of adaptive immunity (lane 253) in skin carcinogenesis is only scarcely explored, thus, probably, does not deserve special discussion.

3)     Minimal editorial work should be done. E.g., in lane 429, instead of  ..BRAF patients…, one could use: … patients with BRAF mutations…?

Conclusion: the review article is well written, providing a rather useful review which may consolidate different studies on the role of gut microbiome in melanoma treatment. Unfortunately, current knowledge does not allow direct correlations between gut microbiota and skin melanoma progression.

The article may be published in the Journal with only minor changes shown under Remarks. Minimal English copy-editing is also required.

The review article concerns rather complex interrelations between gut mictobiota and immune mechanisms of malignancy control. The effects of two different types of immune-targeted drugs, immune checkpoint inhibitors (ICI) and BRAF/MEK inhibitors are highlighted, because of their potential usage for melanoma treatment.

As main item, the role of gut (to lesser extent, skin) microbiomes is discussed, due to its major role in development of the overall antitumor response. The authors suggest dual effects of microbiota, i.e., toxicity and carcinogenicity of some gut bacteria, and, generally, positive effects on antitumor immunity caused by the microbiota, including potential promotion of chemotherapy efficiency.  The main point concerns probable mechanisms of immune therapy by ICI based on clinical data on immunomodulatory effects of Bacteroides and other bacterial classes. Probable manipulation of microbiota by pre- and probiotic therapy or fecal microbiota transplantation (FMT) is considered a promising way to promote antitumor immunity, improve microenviroment and modify clinical responses to ICI therapy in metastatic melanoma. However, further experimental and clinical data are required in the field.

Hence, the article provides certain information concerning interactions between gut microbiome, and ICI, or BRAF-targeted drugs in melanoma patients.

Remarks:

1)     The section Innate Immunity (Lane 211 and further on), concerns the effects of complement activation as a potentially carcinogenic factor in chronic inflammation. In this view, the role of granulocytes as ROI producers could be also mentioned, due to their presence in the inflammatory foci.

2)     Unlike oral or colon cancer, the role of gut microbiota-induced changes of adaptive immunity (lane 253) in skin carcinogenesis is only scarcely explored, thus, probably, does not deserve special discussion.

3)     Minimal editorial work should be done. E.g., in lane 429, instead of  ..BRAF patients…, one could use: … patients with BRAF mutations…?

Conclusion: the review article is well written, providing a rather useful review which may consolidate different studies on the role of gut microbiome in melanoma treatment. Unfortunately, current knowledge does not allow direct correlations between gut microbiota and skin melanoma progression.

The article may be published in the Journal with only minor changes shown under Remarks. Minimal English copy-editing is also required.

The review article concerns rather complex interrelations between gut mictobiota and immune mechanisms of malignancy control. The effects of two different types of immune-targeted drugs, immune checkpoint inhibitors (ICI) and BRAF/MEK inhibitors are highlighted, because of their potential usage for melanoma treatment.

As main item, the role of gut (to lesser extent, skin) microbiomes is discussed, due to its major role in development of the overall antitumor response. The authors suggest dual effects of microbiota, i.e., toxicity and carcinogenicity of some gut bacteria, and, generally, positive effects on antitumor immunity caused by the microbiota, including potential promotion of chemotherapy efficiency.  The main point concerns probable mechanisms of immune therapy by ICI based on clinical data on immunomodulatory effects of Bacteroides and other bacterial classes. Probable manipulation of microbiota by pre- and probiotic therapy or fecal microbiota transplantation (FMT) is considered a promising way to promote antitumor immunity, improve microenviroment and modify clinical responses to ICI therapy in metastatic melanoma. However, further experimental and clinical data are required in the field.

Hence, the article provides certain information concerning interactions between gut microbiome, and ICI, or BRAF-targeted drugs in melanoma patients.

Remarks:

1)     The section Innate Immunity (Lane 211 and further on), concerns the effects of complement activation as a potentially carcinogenic factor in chronic inflammation. In this view, the role of granulocytes as ROI producers could be also mentioned, due to their presence in the inflammatory foci.

2)     Unlike oral or colon cancer, the role of gut microbiota-induced changes of adaptive immunity (lane 253) in skin carcinogenesis is only scarcely explored, thus, probably, does not deserve special discussion.

3)     Minimal editorial work should be done. E.g., in lane 429, instead of  ..BRAF patients…, one could use: … patients with BRAF mutations…?

Conclusion: the review article is well written, providing a rather useful review which may consolidate different studies on the role of gut microbiome in melanoma treatment. Unfortunately, current knowledge does not allow direct correlations between gut microbiota and skin melanoma progression.

The article may be published in the Journal with only minor changes shown under Remarks. Minimal English copy-editing is also required.

The review article concerns rather complex interrelations between gut mictobiota and immune mechanisms of malignancy control. The effects of two different types of immune-targeted drugs, immune checkpoint inhibitors (ICI) and BRAF/MEK inhibitors are highlighted, because of their potential usage for melanoma treatment.

As main item, the role of gut (to lesser extent, skin) microbiomes is discussed, due to its major role in development of the overall antitumor response. The authors suggest dual effects of microbiota, i.e., toxicity and carcinogenicity of some gut bacteria, and, generally, positive effects on antitumor immunity caused by the microbiota, including potential promotion of chemotherapy efficiency.  The main point concerns probable mechanisms of immune therapy by ICI based on clinical data on immunomodulatory effects of Bacteroides and other bacterial classes. Probable manipulation of microbiota by pre- and probiotic therapy or fecal microbiota transplantation (FMT) is considered a promising way to promote antitumor immunity, improve microenviroment and modify clinical responses to ICI therapy in metastatic melanoma. However, further experimental and clinical data are required in the field.

Hence, the article provides certain information concerning interactions between gut microbiome, and ICI, or BRAF-targeted drugs in melanoma patients.

Remarks:

1)     The section Innate Immunity (Lane 211 and further on), concerns the effects of complement activation as a potentially carcinogenic factor in chronic inflammation. In this view, the role of granulocytes as ROI producers could be also mentioned, due to their presence in the inflammatory foci.

2)     Unlike oral or colon cancer, the role of gut microbiota-induced changes of adaptive immunity (lane 253) in skin carcinogenesis is only scarcely explored, thus, probably, does not deserve special discussion.

3)     Minimal editorial work should be done. E.g., in lane 429, instead of  ..BRAF patients…, one could use: … patients with BRAF mutations…?

Conclusion: the review article is well written, providing a rather useful review which may consolidate different studies on the role of gut microbiome in melanoma treatment. Unfortunately, current knowledge does not allow direct correlations between gut microbiota and skin melanoma progression.

The article may be published in the Journal with only minor changes shown under Remarks. Minimal English copy-editing is also required.

The review article concerns rather complex interrelations between gut mictobiota and immune mechanisms of malignancy control. The effects of two different types of immune-targeted drugs, immune checkpoint inhibitors (ICI) and BRAF/MEK inhibitors are highlighted, because of their potential usage for melanoma treatment.

As main item, the role of gut (to lesser extent, skin) microbiomes is discussed, due to its major role in development of the overall antitumor response. The authors suggest dual effects of microbiota, i.e., toxicity and carcinogenicity of some gut bacteria, and, generally, positive effects on antitumor immunity caused by the microbiota, including potential promotion of chemotherapy efficiency.  The main point concerns probable mechanisms of immune therapy by ICI based on clinical data on immunomodulatory effects of Bacteroides and other bacterial classes. Probable manipulation of microbiota by pre- and probiotic therapy or fecal microbiota transplantation (FMT) is considered a promising way to promote antitumor immunity, improve microenviroment and modify clinical responses to ICI therapy in metastatic melanoma. However, further experimental and clinical data are required in the field.

Hence, the article provides certain information concerning interactions between gut microbiome, and ICI, or BRAF-targeted drugs in melanoma patients.

Remarks:

1)     The section Innate Immunity (Lane 211 and further on), concerns the effects of complement activation as a potentially carcinogenic factor in chronic inflammation. In this view, the role of granulocytes as ROI producers could be also mentioned, due to their presence in the inflammatory foci.

2)     Unlike oral or colon cancer, the role of gut microbiota-induced changes of adaptive immunity (lane 253) in skin carcinogenesis is only scarcely explored, thus, probably, does not deserve special discussion.

3)     Minimal editorial work should be done. E.g., in lane 429, instead of  ..BRAF patients…, one could use: … patients with BRAF mutations…?

Conclusion: the review article is well written, providing a rather useful review which may consolidate different studies on the role of gut microbiome in melanoma treatment. Unfortunately, current knowledge does not allow direct correlations between gut microbiota and skin melanoma progression.

The article may be published in the Journal with only minor changes shown under Remarks. Minimal English copy-editing is also required.

The review article concerns rather complex interrelations between gut mictobiota and immune mechanisms of malignancy control. The effects of two different types of immune-targeted drugs, immune checkpoint inhibitors (ICI) and BRAF/MEK inhibitors are highlighted, because of their potential usage for melanoma treatment.

As main item, the role of gut (to lesser extent, skin) microbiomes is discussed, due to its major role in development of the overall antitumor response. The authors suggest dual effects of microbiota, i.e., toxicity and carcinogenicity of some gut bacteria, and, generally, positive effects on antitumor immunity caused by the microbiota, including potential promotion of chemotherapy efficiency.  The main point concerns probable mechanisms of immune therapy by ICI based on clinical data on immunomodulatory effects of Bacteroides and other bacterial classes. Probable manipulation of microbiota by pre- and probiotic therapy or fecal microbiota transplantation (FMT) is considered a promising way to promote antitumor immunity, improve microenviroment and modify clinical responses to ICI therapy in metastatic melanoma. However, further experimental and clinical data are required in the field.

Hence, the article provides certain information concerning interactions between gut microbiome, and ICI, or BRAF-targeted drugs in melanoma patients.

Remarks:

1)     The section Innate Immunity (Lane 211 and further on), concerns the effects of complement activation as a potentially carcinogenic factor in chronic inflammation. In this view, the role of granulocytes as ROI producers could be also mentioned, due to their presence in the inflammatory foci.

2)     Unlike oral or colon cancer, the role of gut microbiota-induced changes of adaptive immunity (lane 253) in skin carcinogenesis is only scarcely explored, thus, probably, does not deserve special discussion.

3)     Minimal editorial work should be done. E.g., in lane 429, instead of  ..BRAF patients…, one could use: … patients with BRAF mutations…?

Conclusion: the review article is well written, providing a rather useful review which may consolidate different studies on the role of gut microbiome in melanoma treatment. Unfortunately, current knowledge does not allow direct correlations between gut microbiota and skin melanoma progression.

The article may be published in the Journal with only minor changes shown under Remarks. Minimal English copy-editing is also required.

The review article concerns rather complex interrelations between gut mictobiota and immune mechanisms of malignancy control. The effects of two different types of immune-targeted drugs, immune checkpoint inhibitors (ICI) and BRAF/MEK inhibitors are highlighted, because of their potential usage for melanoma treatment.

As main item, the role of gut (to lesser extent, skin) microbiomes is discussed, due to its major role in development of the overall antitumor response. The authors suggest dual effects of microbiota, i.e., toxicity and carcinogenicity of some gut bacteria, and, generally, positive effects on antitumor immunity caused by the microbiota, including potential promotion of chemotherapy efficiency.  The main point concerns probable mechanisms of immune therapy by ICI based on clinical data on immunomodulatory effects of Bacteroides and other bacterial classes. Probable manipulation of microbiota by pre- and probiotic therapy or fecal microbiota transplantation (FMT) is considered a promising way to promote antitumor immunity, improve microenviroment and modify clinical responses to ICI therapy in metastatic melanoma. However, further experimental and clinical data are required in the field.

Hence, the article provides certain information concerning interactions between gut microbiome, and ICI, or BRAF-targeted drugs in melanoma patients.

Remarks:

1)     The section Innate Immunity (Lane 211 and further on), concerns the effects of complement activation as a potentially carcinogenic factor in chronic inflammation. In this view, the role of granulocytes as ROI producers could be also mentioned, due to their presence in the inflammatory foci.

2)     Unlike oral or colon cancer, the role of gut microbiota-induced changes of adaptive immunity (lane 253) in skin carcinogenesis is only scarcely explored, thus, probably, does not deserve special discussion.

3)     Minimal editorial work should be done. E.g., in lane 429, instead of  ..BRAF patients…, one could use: … patients with BRAF mutations…?

Conclusion: the review article is well written, providing a rather useful review which may consolidate different studies on the role of gut microbiome in melanoma treatment. Unfortunately, current knowledge does not allow direct correlations between gut microbiota and skin melanoma progression.

The article may be published in the Journal with only minor changes shown under Remarks. Minimal English copy-editing is also required.

The review article concerns rather complex interrelations between gut mictobiota and immune mechanisms of malignancy control. The effects of two different types of immune-targeted drugs, immune checkpoint inhibitors (ICI) and BRAF/MEK inhibitors are highlighted, because of their potential usage for melanoma treatment.

As main item, the role of gut (to lesser extent, skin) microbiomes is discussed, due to its major role in development of the overall antitumor response. The authors suggest dual effects of microbiota, i.e., toxicity and carcinogenicity of some gut bacteria, and, generally, positive effects on antitumor immunity caused by the microbiota, including potential promotion of chemotherapy efficiency.  The main point concerns probable mechanisms of immune therapy by ICI based on clinical data on immunomodulatory effects of Bacteroides and other bacterial classes. Probable manipulation of microbiota by pre- and probiotic therapy or fecal microbiota transplantation (FMT) is considered a promising way to promote antitumor immunity, improve microenviroment and modify clinical responses to ICI therapy in metastatic melanoma. However, further experimental and clinical data are required in the field.

Hence, the article provides certain information concerning interactions between gut microbiome, and ICI, or BRAF-targeted drugs in melanoma patients.

Remarks:

1)     The section Innate Immunity (Lane 211 and further on), concerns the effects of complement activation as a potentially carcinogenic factor in chronic inflammation. In this view, the role of granulocytes as ROI producers could be also mentioned, due to their presence in the inflammatory foci.

2)     Unlike oral or colon cancer, the role of gut microbiota-induced changes of adaptive immunity (lane 253) in skin carcinogenesis is only scarcely explored, thus, probably, does not deserve special discussion.

3)     Minimal editorial work should be done. E.g., in lane 429, instead of  ..BRAF patients…, one could use: … patients with BRAF mutations…?

Conclusion: the review article is well written, providing a rather useful review which may consolidate different studies on the role of gut microbiome in melanoma treatment. Unfortunately, current knowledge does not allow direct correlations between gut microbiota and skin melanoma progression.

The article may be published in the Journal with only minor changes shown under Remarks. Minimal English copy-editing is also required.
